# Service Orientation and Customer Performance: Triad Perspectives of Sales Managers, Sales Employees, and Customers

**DOI:** 10.3390/bs12100373

**Published:** 2022-09-30

**Authors:** Ho-Taek Yi, MinKyung Lee, Kyungdo Park

**Affiliations:** 1Department of Business Administration, Keimyung University, Daegu 42601, Korea; 2Marketing Department, Sogang University, Seoul 04107, Korea; 3Department of Business Administration, Sogang University, Seoul 04107, Korea

**Keywords:** service orientation, customer orientation, service performance, perceived authenticity, customer loyalty, customer performance, department stores

## Abstract

This study examines how shop managers’ attitudes toward customers are transferred to sales employees, and thus affect customer performance. We surveyed shop managers, sales employees, and customers in five department stores in Seoul, South Korea, in June 2021 to determine the relationships among service orientation, customer orientation, customers’ perceptions of sales employees’ authenticity, and customer performance. We found that sales managers’ service orientation positively influences sales employees’ service and customer orientation. Furthermore, this orientation positively correlates with customers’ perceptions of sales employees’ authenticity, thereby improving service performance and customer loyalty. Few studies have examined how institutional-level capacity and attitudes influence employees within organizations and how this, in turn, affects service performance. Thus, this study theoretically and empirically explores how sales managers’ attitudes and sales behaviors are transferred to sales employees and how this affects customer performance. The research findings fill a gap in the current understanding of customer performance in the service industry.

## 1. Introduction

One of the most significant challenges for retail managers is achieving and maintaining customer satisfaction and loyalty. However, there exist factors that are partially outside the control of retail managers; therefore, the ability to create a satisfactory experience for consumers remains, to a considerable degree, not only in the hands of the management but also in that of the retail staff [1]. Furthermore, competition in the retail service industry makes the quality of service an important determinant in customer satisfaction and customer loyalty [2,3]. The retail industry is a highly competitive business with considerable economic implications, and global retail sales reached over 5.6 trillion USD in 2020. This has forced traditional service industries and department stores to introduce novel sales activities because of greater competition.

Furthermore, retailers demand greater employee performance, especially during sales encounters, to differentiate themselves from competitors and to secure clientele [4,5,6,7]. Consequently, department stores consider sales training as a strategic tool to control and improve the performance of their sales employees, for example, by implementing and monitoring mystery shopper programs [8]. Retailers strive to differentiate themselves from the competition by offering customers a positively distinct shopping experience that will bring them back for more.

Department store sales employees provide physical labor, emotional labor [9], and sales expertise. Thus, the service orientation of both sales employees and managers should be investigated at an individual level, with the expectation that managers’ service attitudes and perceptions directly influence sales employees as the latter interact with customers [10]. Liang et al. [11] found that sales employees’ service orientation affects customers’ service perceptions and attitudes, thereby confirming the importance of employee attitudes and perceptions of a given business. Therefore, understanding the attitude of sales employees toward their work and customers is critical in the department store industry [12].

Given that department store sales employees bridge the gap between sales managers and customers, their performance results from multilateral relationships among this triad of parties.

Although most related studies portray the sales employee–customer relationship as occurring between the seller and buyer [13,14,15], it is important to understand how sales employees obtain service knowledge from sales managers. Internalization of such knowledge can enhance the development of increasingly differentiated service-oriented strategies to help department stores obtain a competitive advantage. However, in the department store industry in South Korea, the contract structure between the department store and manufacturers is becoming an obstacle that prevents them from having a competitive advantage in terms of services. The department store’s purchase method is not “direct buying” but a “consignment contract,” that is, an agreement to claim profit after calculating commission from sales. Under this contract structure, department stores may not do their best to sell, which is why in practice, manufacturers dispatch their sales managers and employees to the department store. This type of sales channel between department stores and manufacturers is called a partially integrated channel (PIC). A PIC is defined as “a single vertical channel structure in which both market governance and hierarchical governance exist; the advantage from a governance perspective is that they encompass the simultaneous employment of multiple governance forms within a single channel” [16] (pp. 603–604). However, PICs also hinder the efficiency of communication between product category managers and frontline employees at the sales point because product category managers are employees of department stores, while sales managers and sales employees are directly employed by manufacturers.

This situation improves personal workplace relationships (PWRs), which are informal, mutual, and unique voluntary interpersonal relationships in which [17] sales managers and sales employees know and communicate with each other as unique individuals and have a relatively strong emotional connection [17]. As they do not belong to the department stores, it is difficult for them to have a formal relationship with the department store’s branch manager, and, thus, personal informal communication between members in the same workplace is voluntarily strengthened. Even if it is difficult to have a role-based relationship within the organization, if the attitude and perception of the sales managers can affect salespeople through PWRs, it can become a major source of competitive advantage that can set a business apart from its competitors [18]. As service professionals are the customers’ primary points of contact, organizations’ service-oriented values and principles directly affect the sales employees’ customer-oriented mindset [19], which is crucial for service organizations. Nonetheless, current research shows that limited empirical studies have investigated the correlation among these concepts, especially in South Korea. Reportedly, no study has established causal relationships between these concepts through methodological input.

To better assess the manager–employee relationship, our study survey included managers and sales employees working in the same section of the department store. The current study provides further insights into the possible correlation between customer orientation and experiences, thereby reducing the standard method bias evident in previous studies on this topic. In this respect, the main objective of this study is to establish and validate a conceptual model that integrates the correlations among service orientation, customer orientation, perceived authenticity, service performance, and customer loyalty within the context of customers’ service perceptions and attitudes in South Korea. As mentioned, most department store businesses in South Korea are based on consignment contracts, wherein stores often employ sales managers and manufacturers employ sales staff. For example, 60,000–70,000 manufacturers’ sales employees work full-time in retail stores, with 200–300 per retail location [20]. Engaged employees are critical to the success of the service industry. This study analyzes multiple respondent sources, including sales managers, sales employees, and customers, to investigate the agents that transfer knowledge and emotions and measure service perceptions and attitudes. It investigates sales employees’ service and customer orientation—filtered through the organizational environment (i.e., managers’ service-oriented attitude) and channeled toward improving service performance—and customer loyalty. Accordingly, the following research questions are raised:Does the service orientation of department store managers affect the individual-level service and customer orientation of sales employees?Do sales employees’ service and customer orientation positively impact customers to perceive service authenticity, thus resulting in effective service performance and customer loyalty?

Additionally, their full-time work at an exchange partner’s facilities would include performing traditional functions in a partially integrated sales channel [21]. What drives their delivery from a store manager to sales employee service orientation and, ultimately, their service orientation and customer orientation? Surprisingly, the current retail literature does not answer these pertinent questions. The literature tends to recommend a study design different from that employed here to examine the associations suggested in the current research model. Notably, most existing studies that assess how customer orientation influences customer experience employ the same population sample. That is, they use only employee or customer responses. Consequently, the methodological input of the current study provides further insight into the possible correlation via face-to-face interviews with sales employees and customers through judgment sampling, thereby reducing the standard method bias evident in previous studies on this topic.

Pre-existing literature has generally focused on individual sales employees’ service capacity and attitudes and not on how institutional-level capacity and attitudes are transferred to individual employees. However, the current study argues that knowledge and emotions are transferrable—sales managers’ attitudes toward customers are transferred to sales staff and, in turn, transmitted to customers, thus affecting sales and service performance. Accordingly, this study fills the gap in the current understanding of customer performance in the service-oriented industry by theoretically and empirically exploring (a) the relationship between the attitudes and sales behaviors of sales managers and employees and (b) how this affects customer performance.

This study theoretically and empirically explores how sales managers’ attitudes and sales behaviors are transferred to employees and how this would affect customer attitudes. Customers’ perceived authenticity of sales employees triggers positive customer behaviors, such as higher customer satisfaction, positive evaluations, repurchasing behaviors, or increased customer loyalty [22,23,24]. Even for a negative situation or negative emotions, sales employees’ authentic service attitudes can transform a customer’s negative emotional state into a more positive one [25].

The remainder of this study is structured as follows. Section 2 provides an overview of the relevant literature regarding the constructs used in this study: service orientation, customer orientation, perceived authenticity, customer performance, and customer loyalty. Furthermore, the hypotheses proposed in this study are described. Section 3 presents the materials and methods, and Section 4 discusses the hypotheses testing and the findings. Section 5 concludes with a discussion of their implications and limitations.

## 2. Literature Overview and Derivation of Hypotheses

### 2.1. Service Orientation

*Service orientation* refers to the managerial philosophy implied by policies, procedures, and goals [26]. Some scholars understand it as a way of practicing customer-oriented marketing strategies [27,28,29], the definition of service orientation varying in the literature, although most studies refer to it as “an attitude of an employee to satisfy the service needs of the customers” [30].

Service orientation is the ongoing organizational process of supporting and providing services intended to create service excellence [31] or implementing customer-oriented marketing strategies [32]. It improves customer satisfaction by prioritizing value creation and excellence in service delivery [33,34]. A service-oriented organizational culture has a strong positive effect that is transferred to the service mindset of sales employees and helps achieve competitive advantages [35]. Moreover, highly service-oriented employees can positively affect other employees’ attitudes and performance [36,37]. Prior studies have focused on examining the effect of service orientation at individual and organizational levels [38] and explaining organizational factors that foster employees’ service orientation [39,40]. This process is considered an antecedent to higher job satisfaction [41], customer satisfaction [42], and sales employees’ job security [43]. Furthermore, it improves employees’ task performance, resulting in greater organizational sales performance [44,45]. Employees’ service orientation can be fostered through institutional support [46], such as training [47,48], recognition [49], and compensation [50]. Li et al. [51] found that employees with high outcome expectancies and instrumentalities (i.e., expectations that their efforts will result in a greater reward) are more likely to exhibit a greater level of service orientation. Working in an organization with a service-oriented culture encourages sales employees to establish long-term relationships with customers, and to expect that their efforts are valued by the organization [46,52]. Based on this understanding of service orientation, the current study examines how sales employees’ service orientation is channeled from managers to customers by focusing on individual departments within the department store.

### 2.2. Customer Orientation

Research on customer orientation has consistently attracted the attention of both scholars and managers, as customer-oriented value creation is a critical component of sustainable organizational development. Saxe and Weitz [53] were the first to suggest the concept of customer orientation and defined it as “the degree to which salespeople practice marketing concepts by trying to help their customers make purchase decisions that will satisfy their needs” [53] (p. 344). Therefore, service orientation can be considered an organizational activity that involves generating and delivering a quality service for organizational survival and profit generation, while customer orientation can be considered an effort toward achieving service orientation. Customer orientation thus emphasizes organizational and employees’ efforts to think from the perspective of customers and perform subsequent organizational activities to fulfill their needs and ensure their satisfaction, which would likely result in profit generation and organizational survival. Studies on customer orientation have consistently attracted the attention of both scholars and managers, as customer-oriented value creation is a critical component of sustainable organizational development. Jarideh [54] noted that customer orientation refers to customer knowledge and the ability to create superior value consistently. Customer orientation can be defined as the employee’s inclination or tendency to meet consumer needs as part of their job responsibilities [55]. Customer orientation is also defined as the firm’s ability and willingness to identify, assess, comprehend, and respond to consumer needs [56]. Therefore, while service orientation is an organizational activity involving the production and delivery of quality service for organizational survival and profit generation, customer orientation emphasizes that organizations should consider customers’ perspectives and perform subsequent organizational activities to satisfy their customers [57]. This customer focus likely results in profit generation and organizational survival [58].

While service orientation implies proactive behavior to provide excellent service to clients, a customer orientation approach targets a steady outcome of customer satisfaction by pursuing a highly gratifying service for customers [59]. To possess high customer orientation, organizations need to understand their targeted customers properly [60]. Therefore, maintaining an organizational culture that emphasizes customer values and needs and a support system is critical to serving customers’ needs better. Moreover, employees’ perceptions of the organizational support they receive is more likely to instill emotional attachment toward the organization [61], thereby motivating them to make greater efforts to achieve organizational goals in terms of customer orientation.

### 2.3. Perceived Authenticity

*Authenticity* has been extensively studied in the field of psychology. Recently, this concept has also been attracting attention in marketing research, as customers’ awareness of the moral values of corporate activities has increased. Harter [62] defined authenticity as “owning one’s personal experiences, thoughts, emotions, needs, wants, preferences, or beliefs…act[ing] in accordance with the true self, [and] expressing oneself in ways that are consistent with inner thoughts and feelings” (p. 382). Customers’ perception of authenticity at the point of contact refers to their emotional reaction to the sales employees’ perceived kindness and friendliness or passion for service excellence [63]. Authenticity can be understood using concepts such as the emotional contagion phenomenon. The emotional contagion theory describes that nonverbal cues such as a smile or glance during interaction may not only lead to exchange of emotions and feelings between individuals but also influence their emotional states [64]. Therefore, customers can perceive authenticity through their interactions with salespeople at the point of contact, but salespeople can be in turn affected by a positive statement from customers. Perceived authenticity helps customers evaluate services, products, brands, and companies [38,65,66,67,68,69,70]. Therefore, companies must care about product and service quality and ensure that customers perceive authenticity to gain comparative advantages [71,72]. Researchers have found that consumers are motivated to focus on symbolic cues that convey product or brand authenticity [67,72]. Kammeyer-Mueller et al. [73] found that customers’ perceived authenticity of sales employees triggers positive customer behaviors, such as higher customer satisfaction, positive evaluations, repurchasing behaviors, and increased customer loyalty [74]. Even when negative emotions have been aroused, sales employees’ authentic service attitudes can transform a customer’s negative emotional state into a positive one [75]. In other words, it is necessary to provide authentic services or to meet the customers’ desired needs and service level. Thus, this study adopts Bae’s [76] definition, referring to perceived authenticity as the customers’ perceptions that are developed through interactions with sales employees—it is important to understand how sales staff’s authenticity is acquired and delivered to customers.

### 2.4. Understanding Customer Performance: Service Performance and Customer Loyalty

As the managerial practices of the service industry develop, the conceptual definition of service performance has also evolved. For instance, Miller (1990) defined service performance as the result of employees’ efforts to fulfill organizational goals and objectives. Zeithaml [77] described it as receiving positive customer evaluations on overall service quality, while Swan and Combs [78] categorized service performance into two subcategories—instrumental and expressed performance.

Service performance is evaluated based on organizational objectives and goals [79]. Particularly for service organizations, service performance refers to the process of service delivery wherein providers respond to and serve customers [80]. Organizational infrastructure relationships are characterized by contact and communication occurring in the process of performing one’s job or performing a given organizational role [18]. Netemeyer and Maxham [81] defined service performance as the result of employees’ efforts to fulfill organizational goals and objectives, while Roy et al. [82] described it as receiving positive customer evaluations on overall service quality. A common approach to capture employees’ service performance entails analyzing customer evaluations of service quality and the overall interaction [83,84]. Several empirical studies have examined the service performance of sales employees or managers. They show that service attitudes such as high customer orientation are more likely to induce long-term customer satisfaction [85,86] and excellent sales performance [87]. As employees’ attitudes play an important role in customers’ perceptions of products and services, they can help the management succeed and raise customers’ service quality evaluations [84,88]. Therefore, better service performance can increase market share and profitability through a price premium [88], thereby rewarding organizations that focus on improving service performance.

### 2.5. Customer Loyalty

Customer loyalty goes beyond personal preference, as it entails expanded interest in a brand and an organization, such that customers act as facilitators of organizational marketing practices. Other than the evaluations of service performance, another aspect of customers’ response to service provision is exhibited customer loyalty [89,90,91]. Critical determinants of a service organization’s long-term success are developing, maintaining, and enhancing customer loyalty by minimizing churn, which is defined as lost customers [89]. As such, customer loyalty is a critical indicator because it captures the quality of the relationship between customers and a firm [90,91] and customers’ intention to continue engaging with a service provider based on their experiences and expectations [92,93]. Therefore, with an appreciation of the importance of customer loyalty for sustainable success in a service-oriented organization, this study analyzes how customer loyalty can be achieved, primarily through employees’ service- and customer-oriented attitudes. The current study thus investigates how managers’ individual-level service orientation affects sales employees’ service and customer orientation when working at stores.

### 2.6. Research Model

Figure 1 presents a graphical representation of this study. We propose that the sales manager’s service orientation is transferred to sales employees’ attitudes toward service and customer orientation. Sales employees’ customer and service orientation then affects customers’ performance and perceptions of authenticity, such as service performance and customer loyalty.

### 2.7. Hypotheses

#### 2.7.1. Relationship between Sales Managers’ and Employees’ Service Attitudes

Customers need positive service encounters to be satisfied with service quality [82,85]. Thus, understanding how sales employees exhibit and instill good service attitudes is of great interest to service-oriented organizations. The major influencing factors are the attitudes and behaviors of managers who work closely with the employees on site. Managers often try to establish a particular service environment at their stores.

First, they set and communicate a clear vision to their employees [94]. The top-down approach of delivering the service vision would likely affect employees who exhibit similar service attitudes. Managers with high service orientation have a clear vision of the organization’s service objectives and goals, reinforcing the importance of service quality and customer satisfaction [42]. Reiterating the service vision and objectives during daily meetings or addressing customer complaints makes employees more likely to internalize the vision as specified by service-oriented managers [38]. Therefore, sales employees are more likely to exhibit high service attitudes aligned with their managers’ communicated vision.

Second, managers establish a positive service environment through their exemplary attitudes and behaviors [95]. Managers with high service orientation are more likely to emphasize service quality and the fulfillment of the needs of customers, the recipients of service activities [96], as they believe in the importance of service quality and customer satisfaction [42]. Employees in the same unit would, directly and indirectly, observe and emulate such attitudes and behaviors. Furthermore, service-oriented managers are more likely to be motivated to train employees to maximize customer satisfaction. Suppose employees receive excellent training from their managers. In that case, they are more likely to exhibit a high commitment to their work, have job satisfaction [97,98], and provide excellent service to customers [99]. Rather than directing and shaping the service climate by simply dictating service-oriented attitudes, service-oriented managers become an example of how sales employees need to behave during sales encounters [100,101]. The emotional contagion phenomenon can also explain the influence of sales managers’ attitudes on sales employees: Nonverbal cues such as smiles and glances are transferred to others and influence their emotional states during interactions [102,103]. Managers with high service orientation often exhibit positive energy and emotions when interacting with customers and employees. This positivity is subsequently transferred to employees and contributes to creating a more pleasant work environment. Hence, the organizational environment positively influences organizational members, resulting in a well-functioning organization [18].

Yoon et al. [104] found that the organizational climate contributes directly to employees’ work effort and job satisfaction and indirectly impacts customers’ perceptions of employees’ service quality. Thus, combined with the emotional contagion theory [105], managers’ positive attitudes can be transferred to employees and customers, thus naturally yielding more positive service attitudes among employees toward customers. Customer performance is closely related to the service orientation of an organization, including the organization’s service vision and system, thus encouraging employees to deliver authentic services. Therefore, employees add value to the corporate performance by performing customer orientation roles when they are satisfied with their jobs and immersed in the organization. This leads to our first two hypotheses:

**H1.** *Sales managers’ and sales employees’ service orientation are positively related*.

**H2.** *Sales managers’ and sales employees’ customer orientation are positively related*.

#### 2.7.2. Sales Employees’ Service and Customer Orientation and Perceived Authenticity

To ensure that customers perceive authenticity during service encounters, sales employees should understand perception sources—when and at which point customers perceive authenticity and how sales employees can exhibit such elements. Generally, customers perceive it when sales employees exhibit customer-care behaviors and when the latter’s actions speak of a dedicated effort to consider the customers’ point of view rather than, for instance, their own sales record. Customer-care behaviors are closely related to service orientation, whereas customer benefits are rooted in customer orientation. Customers tend to evaluate service quality positively when sales employees exude positive energy, show emotional expressions, and use supportive words [106]. Therefore, when sales employees provide service and customer orientation, customers are more likely to perceive an authentic relationship. Professional service knowledge is also an important service delivery component, as it contributes to trust and perceived authenticity [107,108]. A sales employee with a high level of service and customer orientation tries to assess customers’ needs and preferences and builds on their professional service knowledge and competence to add value to their service [109]. Li et al. [51] found that sales employees’ customer-oriented behaviors offer more diverse options to provide customers with better service experiences. Such behaviors should also include good rapport and careful listening techniques to identify and understand customers’ needs. They result in effective service performance and customer satisfaction and retention [110].

Furthermore, Coulter and Coulter [107] found that service employees’ competencies and similarities with customers could reduce interpersonal barriers, raise comfort levels, and establish trust. The reduced barriers and feelings of comfort during service encounters enhance perceived authenticity [111]. Thus, personalized service raises customer trust rather than manual or routine services based on substantial professional sales knowledge [106]. Therefore, the following hypotheses are suggested:

**H3.** *Sales employees’ service orientation is positively correlated with customers’ perceptions of authenticity*.

**H4.** *Sales employees’ customer orientation is positively correlated with customers’ perceptions of authenticity*.

#### 2.7.3. Perceived Authenticity and Customer Performance

When customers perceive sales employees’ attitudes and activities as authentic rather than calculated or mechanical, they are more likely to associate positive emotions with the products and services acquired. In addition, Olk et al. [111] showed that the experiential aspect of authenticity, revealed through customers’ joy, happiness, and cultural experiences in consuming products and services, induces brand loyalty [112,113]. Grandey et al. [106] studied the relationship of authenticity with customer satisfaction and rapport with sales employees. They found that when a positive expression embodies authenticity, it increases customer loyalty and contributes to establishing a connection with sales employees, ultimately leading to repurchase intention. Tjahjaningsih et al. [114] found that customer satisfaction and positive affection increased loyalty through patronage or word-of-mouth behaviors. Thus, employees’ authenticity improves customers’ intention to build a long-term relationship with the business and induces positive customer behavior.

Moreover, customers’ perceived authenticity of sales employees’ attitudes induces positive emotions with the products and services received. The positive emotions and attachment further lead to greater commitment to the products, services, or brand, yielding customer loyalty. Focusing on peer-to-peer accommodation services in the US and Finland, Tussyadiah found that travelers’ desire for interactions with locals and authentic experiences changes their traveling behavior [115].

Personal workplace relationships (PWRs) have a stronger emotional component than other workplace relationships, are consensual and mutual, and vary in closeness [18,116]. Sustainable long-term relationships based on extensive interactions between salespeople and customers may eventually lead to salesperson-customer PWRs. In addition, PWRs can affect interpersonal satisfaction and increase intimacy [18].

Similarly, trust in service providers and service quality affirms customers’ needs and desires and enhances brand commitment. This leads to positive purchasing intention and behaviors [117,118]. According to Xiang and Li [119], consumers’ perceived authenticity relating to the quality of the product or service can potentially influence consumers’ purchasing decisions and lower uncertainty about the product. In other words, the service provider does not pretend to be authentic but naturally expresses and delivers to the customer. It is thus important to allow the customer to perceive authenticity in purchasing a product or service.

Therefore, customers’ perceived authenticity of sales employees needs careful examination as a potential contributing factor of the service-oriented organization’s brand management. This leads to the following hypotheses:

**H5.** *Customers’ perceived authenticity of sales employees is positively correlated with service performance*.

**H6.** *Customers’ perceived authenticity of sales employees is positively correlated with customer loyalty*.

## 3. Materials and Methods

### 3.1. Measures

All scales were borrowed or adapted from prior studies and measured on a 7-point Likert scale. This study used four items adapted from Keillor et al. [120] to measure service orientation. The scale was modified to reflect the perspectives of department store managers and sales employees. *Sales employee customer orientation*, measured using five items adapted from Saxe and Weitz [53], reflected employees’ desire to connect with customers and understand the importance of customer orientation for their own performance [121]. *Perceived authenticity* of sales employees was measured using the three items adapted by Yoo and Arnold [122] from the original scale [123]. To measure sales employees’ *service performance*, we used a composite of the empathy and excellent job performance scales, which represented sales employees’ expected behaviors. Empathy comprised two items based on the SERVQUAL empathy scale [124]. A further two-item scale based on the service provider performance scale [125] assessed sales employees’ excellence in performance. *Customer loyalty* was assessed with three items that measured the likelihood of customers returning to purchase additional services or engaging in positive word-of-mouth behaviors. Finally, an adaptation of the original Salanova et al. [126] scale by Swan and Oliver [127] was used to measure the reciprocal relationships between service orientations and customer loyalty. As all the measures were originally written in English, we followed the back-translation procedure recommended by Brislin [128]. First, the English scales were translated into Korean; then, two bilingual experts back-translated the Korean scales into English to minimize translation bias. The measurement items for all constructs are shown in Appendix A.

### 3.2. Sampling Procedure and Data Collection

Since we needed three response sources, this study required the cooperation of the department store headquarters, sales manager, sales employees, and their customers. To test the research hypotheses, we surveyed sales managers, sales employees, and their customers from five department store branches in Seoul, South Korea. Data collection occurred through face-to-face interviews and talks, where the interviewer wrote down the answers on paper, and a paper questionnaire was also distributed so that respondents could read and write down their answers. We conducted the face-to-face interviews with sales employees and customers through judgment sampling (i.e., purposive sampling), a non-probability sampling technique in which responses are chosen based on the researcher’s judgment. To measure the relationship between managers and employees, a paper questionnaire was distributed to sales managers and sales employees working at the same shop. Therefore, with the cooperation of the department store headquarters, sales managers were asked to respond to the questionnaire survey by considering the company (manufacturer) that dispatched them, while the sales employee was asked to answer keeping in mind the company (manufacturer or HR dispatching company) to which they belonged. Moreover, customers were asked by the interviewer to rate the employee–customer relationship on the spot. The interviewer randomly selected two customers who visited each shop in the department store, guided them to a separate location, and conducted the interview and survey; the average of the responses to each question was used. The sample shops included those selling men’s and women’s fashion wear, golf and outdoor goods, and cosmetics. In short, the interviews and survey were conducted with a shop manager, a sales employee, and two customers per shop and targeted 200 shops in department stores (see Table 1 for demographic details). Furthermore, the customer survey was conducted in a location away from the shop with the cooperation of the department store’s head office.

We assessed the possible nonresponse bias in two ways. First, we compared early and late respondents [129] based on the study’s key variables: manager’s service orientation, sales employee’s service orientation, sales employee’s customer orientation, customer’s perceived authenticity of the sales staff, service performance, and customer loyalty. The results indicated the validity of the variables (*p* > 0.05). Second, we compared the response values for each department store branch. No significant differences were found (*p* > 0.05).

### 3.3. Validity and Reliability

The proposed structural equation model was tested using AMOS 18.0. Following the two-step approach suggested by Anderson and Gerbing [130], the measurement validity of each construct was tested before estimating the structural paths to test the hypothesized relationships. We evaluated discriminant validity by performing a confirmatory factor analysis (CFA) consistent with Anderson and Gerbing’s study [130]. We first examined the fit indices of the proposed six-factor model (i.e., managers’ service orientation, sales employees’ service orientation, sales employees’ customer orientation, perceived authenticity, service performance, and customer loyalty). The CFA with the six-factor model showed that the data were a good fit: χ^2^ = 292.253, degrees of freedom (df) = 207, χ^2^/df = 1.412, root mean square error of approximation (RMSEA) = 0.046, goodness-of-fit index (GFI) = 0.886, normal fit index (NFI) = 0.933, relative fit index (RFI) = 0.918, incremental fit index (IFI) = 0.979, Tucker–Lewis index (TLI) = 0.975, and comparative fit index (CFI) = 0.979. As shown in Table 2, the average variance extracted (AVE) of all the constructs was above 0.5, with composite reliability scores greater than 0.7. These results provide evidence supporting convergent validity.

However, as perceived authenticity was highly correlated with customer loyalty (r = 0.708, *p* < 0.01), the discriminant validity between the two variables was still questionable. We performed a CFA with an alternative five-factor model to address this concern, where perceived authenticity and customer loyalty were combined into a single factor. The results showed that the five-factor model’s fit indices (i.e., χ^2^ = 455.845, df = 212, χ^2^/df = 2.150, RMSEA = 0.077, GFI = 0.828, NFI = 0.895, RFI = 0.875, IFI = 0.941, TLI = 0.929, and CFI = 0.940) were worse than those of the six-factor model. The chi-square difference tests revealed that the six-factor model yielded a better fit than the five-factor model (Δχ^2^_(5)_ = 163.592, *p* < 0.01) (In this case, the following holds: (1) the null hypothesis is rejected (and the alternative hypothesis is accepted) if it is difficult for the test statistic to follow the chi-square distribution; (2) if the test statistic can sufficiently occur x, the null hypothesis is rejected, and the confidence level or *p* value is used to determine whether an event is probable or unlikely). This suggested that perceived authenticity and customer loyalty could be separated into two distinct constructs. Moreover, we examined discriminant validity by comparing the AVE with its shared variance with any of the constructs. Fornell and Larcker [131] suggested a higher AVE for each construct than the squared correlation of any other construct. As shown in Table 3, the AVE was greater than all corresponding correlations, which indicated adequate discriminant validity.

## 4. Results

### 4.1. Hypothesis Testing

As shown in Table 4, the structural model’s results indicated the following fit indices: χ^2^ = 368.699, df = 237, χ^2^/df = 1.556, RMSEA = 0.0054, GFI = 0.864, NFI = 0.913, RFI = 0.899, IFI = 0.967, TLI = 0.961, and CFI = 0.967. The structural equation model’s adequacy was evaluated based on the criteria of overall fit with the data.

As revealed in Table 4, all the hypotheses were supported. As expected, shop managers’ service orientation was positively related to sales employees’ service orientation (β = 0.166, *p* < 0.05) and customer orientation (β = 0.297, *p* < 0.01), confirming H1 and H2. H3 and H4 were also supported: customers’ perceived authenticity was greater with higher levels of sales employees’ service orientation (β = 0.194, *p* < 0.01) and customer orientation (β = 0.191, *p* < 0.05). Lastly, perceived authenticity was positively related to service performance (β = 0.665, *p* < 0.01) and customer loyalty (β = 0.753, *p* < 0.01); thus, H5 and H6 were supported. We included the control variable, customers’ shop transaction period, with service performance (β = 0.049, *p* > 0.05) and customer loyalty (β = −0.008, *p* > 0.05).

### 4.2. Additional Analysis

We also investigated the direct effect of managers’ service orientation on customer performance and the direct effect of sales employees’ customer orientation and service orientation on customer performance. Although these variables were not hypothesized in this study, there could be a causal relationship. First, managers’ service orientation had a significant positive effect on service performance (β = 0.122, *p* < 0.05) but not on customer loyalty (β = 0.005, *p* > 0.05). Second, sales employees’ customer orientation had a positive effect on service performance (β = 0.146, *p* < 0.05) but not on customer loyalty (β = −0011, *p* > 0.05). Third, sales employees’ service orientation affected only service performance (β = 0.125, *p* < 0.05) but not customer loyalty (β = 0.060, *p* > 0.05).

## 5. Discussion

Department store sales employees are required to meet organizational requirements and satisfy customer needs. Their performance can be viewed as a function of the manager–sales employee relationship and vice versa. Although studies on department store sales employees and customers have mainly been conducted from the seller–buyer perspective, the relationships between sales employees and customers embody the sales knowledge that employees receive from their managers and adopt in their own behavior. This study is based on the premise that knowledge and emotions are transferrable. It aims to clarify that sales managers’ attitudes toward customers are transferred to the sales staff, whose attitudes in turn are transferred to the customers, thus affecting service performance. It is important to understand the relationship between sales employees’ attitudes and verbal and nonverbal communication, as well as the way customers perceive authenticity, as the perceived authenticity of salespeople increases sales performance [132]. Customers perceive authenticity differently in diverse situations and environments and use various situational cues to evaluate the authenticity of objects based on their knowledge and interest pertaining to specific objects [133]. Furthermore, this study provides empirical evidence that store managers’ attitudes toward customers can be transmitted to sales employees and customers. Several key findings emerge.

First, while previous studies have focused on individual sales employees, this study contributes to the understanding of sales employees’ capabilities and attitudes by focusing on how organizations or people within organizations transfer high-level capabilities and attitudes to individuals or staff at lower levels. Specifically, we examine how the service orientation of shop managers—who interact daily with the store’s sales employees and act as a bridge between them and the organization—affects their employees’ sales orientation and attitudes. We find that shop managers’ service orientation positively affects sales employees’ service and customer orientation.

Second, sales employees’ service and customer orientation affect customers’ evaluation of their authenticity. This result implies that although providing reasonable service to customers is essential, it is of greater significance when customers evaluate its authenticity. Therefore, managers must understand the importance of service and customer orientation and train sales employees accordingly.

Third, customers’ perceptions of the authenticity of services affect customer performance and loyalty and employees’ service performance. The empirical findings suggest that securing customers’ perceptions of authenticity is imperative because perceived authenticity mediates the relationship between managers’ or sales employees’ service orientation and customer loyalty. 

Fourth, the organizational environment is influenced by tangible relationships, meaning that an environment in which members are balanced in relationships can be a positive influence and function well [133,134]. This environment is transferred to the customer by the salesperson, facilitating the development of a personal workplace relationship (PWR), and the degree of PWRs increases intimacy, which can in turn affect work performance [18]. Conversely, a poor relationship can lead to negative ripple effects and mistrust, which can ultimately end the relationship. In addition, PWRs have a negative side that can lead to inappropriate behavior in the organization due to the inappropriate retention of relationships. Salespeople know that the strength and depth of their workplace relationships have a significant impact on the service industry, and the discussion of these relationships is consistent with the definition of PWRs [18].

The results suggest that sales employees’ service attitude affects customer performance. Therefore, sales employees’ attitudes toward services and customers are fundamental for a service-oriented organization to establish sustainable long-term relationships with customers. Previous studies have underscored the importance of sales employees. These findings are relevant because sales employees interact with customers and represent the organization during sales encounters [135,136,137]. This study is unique because it measures the variables used in the research model from the perspective of respondents such as shop managers, sales employees, and customers in the store. A triad perspective is thus established to portray the dynamic yet intricate strategies that the three parties implement.

### 5.1. Managerial Implications

The current study’s empirical findings provide practical and managerial implications for sales organizations, especially large retailers and department stores. An appropriate education and compensation system should be introduced to effectively manage customer orientation. An atmosphere allowing active management support and the free exchange of opinions should be provided. The study suggests adopting a method based on organizational goals and objectives to achieve customer performance.

First, the findings highlight why managers should cultivate service orientation: A manager’s service orientation impacts sales employees and, ultimately, customer performance. Sales managers are expected to increase the sales employees’ service orientation by strengthening workplace relationships. In role-based relationships, sales managers need to communicate in the sales process, emphasizing the goal of the sales organization, that is, service orientation, and the virtues that sales employees are expected to possess. In PWRs, sales managers become role models so that the sales employees can voluntarily show service-oriented behaviors in the sales process. Therefore, sales managers need to not only simply communicate as role occupants to accomplish work tasks [16,138] but also share a unique experience with sales employees to secure emotional ties or personal intimacy. Moreover, individual service orientation can be disseminated as organizational culture because lower-level employees interact frequently and directly with managers. Service organizations should pay specific attention to and invest in providing customer service-related training to instill awareness of the importance of service-provisioning attitudes and behaviors in store managers and employees to ensure their desired service performance. Therefore, sales organizations need to support service orientation at the firm level.

Second, this study demonstrates the top-to-bottom transferability of service attitudes in practice, allowing for better customer performance in a department store. Examining the transfer of service attitudes can provide managers with valuable insights into designing training mechanisms that instill service attitudes in employees at different levels. This is especially important in South Korea, as product category managers are employed by department stores, whereas manufacturers or tenants appoint sales managers and sales employees. In some cases, sales employees are hired by the manufacturers, but there are also cases where they are hired by an HR outsourcing company, making it difficult for the sales managers to directly train them. Under this fragmented human resource management structure, it is difficult to provide comprehensive sales and service training or sales instruction. Therefore, from the manufacturer’s perspective, it is more appropriate for sales managers to provide implicit rather than explicit training to inculcate service attitude in employees, and as we mentioned above, strengthening the PWR between sales managers and sales employees can be an effective alternative.

Third, sales organizations must instill service and customer orientation to achieve their service objectives and goals. This study reveals that service and customer orientations positively affect service performance. Previous studies have shown that sales employees’ service and customer orientation can be enhanced by their positive attributes [30] and organizational support [139,140].

Fourth, sales organizations should ensure that their employees have perceived authenticity to maintain customer loyalty. As illustrated, businesses can only obtain customer loyalty when perceived authenticity is guaranteed. According to Grandey et al. [132], customers perceive authenticity in sales employees’ attitudes based on verbal and nonverbal communication. Therefore, sales organizations should consider the importance of gestures, eye contact, tone, facial expressions, and verbal communication and train employees to deliver authenticity during sales encounters.

Finally, to establish a service-oriented organizational culture and develop a sincere workforce, managers should focus on developing relationships with their sales employees. Establishing an organizational culture that faithfully preserves the essence and form of attachment to the work and organization is important. This suggests that customer-oriented strategies should be established, keeping in mind employee satisfaction and emotional attachment to the organization. In other words, to raise the level of customer service, it is necessary to establish a desirable organizational culture and develop human resources and internal marketing policies that increase organizational attachment. An organizational culture that focuses on building relationships rather than just performance induces strong employee attachment to the job and organization, positively affecting performance through voluntary efforts to improve customer experience.

### 5.2. Limitations and Future Research

Despite the novel findings on the direct and mediating effects of managers’ and sales employees’ attitudes and orientations, some limitations of this study must be acknowledged. First, although managers, sales employees, and customers were multisource respondents, the survey was conducted at only five department stores. Future studies could expand the data set by engaging different department stores to account for various organizational environments.

Given that the data were obtained from only one service sector, generalizing the outcomes to other service industries should be approached with caution. Moreover, the study sample was specific and did not represent all customers in South Korea and worldwide. Considering these aspects and the fact that this study did not consider cultural factors, future studies should investigate other industries and nations to strengthen the generalizability of the results. In addition, business etiquette, which varies across countries, should be considered.

Second, service orientation is a managerial factor that affects sales employees’ service and customer orientation. However, other managerial and organizational factors, such as store managers’ customer orientation or the organizational control system in service delivery, can also affect sales employees’ perception of service attitudes. Therefore, research is required on the relationship between managers and sales employees and various methodological approaches or insights. This can help organizations influence the way customers perceive and evaluate the services they receive. Moreover, the suggested model in this study is quite simple, as it examines the influence of service orientation based entirely on the performance of customers through the arbitrating role of consumer experience. Therefore, future studies should build more sophisticated service and consumer orientation models.

Finally, as the current study’s service measures were developed relatively recently and have not been used widely, further studies should examine the reliability and validity of service orientation measures in the retail context.

## Figures and Tables

**Figure 1 behavsci-12-00373-f001:**
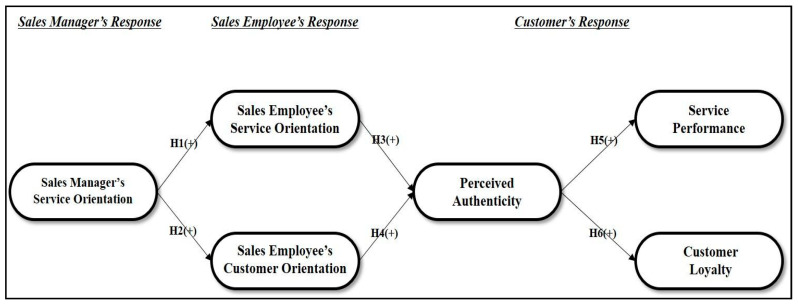
Research model.

**Table 1 behavsci-12-00373-t001:** Sample demographics.

Classification		N	%	Classification		N	%
Department store branch	Trade center	39	20.1	Product category	Golf and outdoor	48	24.7
	Mokdong	39	20.1		Cosmetics	48	24.7
	Pankyo	37	19.1		Men’s fashion	48	24.7
	Cheonho	40	20.6		Women’s fashion	50	25.8
	Shinchon	39	20.1	Manager’s work experience	Under 3 years	99	51.0
Manager’s gender	Man	49	25.3		3–6 years	36	18.6
	Woman	145	74.7		6–9 years	17	8.8
Sales employee’s gender	Man	41	21.1		Over 9 years	42	21.6
	Woman	153	78.8	Department store transaction periods of customers	Under 3 years	83	42.8
Customer’s gender	Man	87	22.4		3–6 years	45	23.2
	Woman	301	77.6		6–9 years	30	15.5
Sales employee’s work experience	Under 3 years	137	70.6		Over 9 years	36	18.6
	3–6 years	40	20.6	Shop transaction periods of customers	Under 3 years	125	64.4
					3–6 years	34	17.5
	Over 6 years	17	8.8		6–9 years	20	10.3
					Over 9 years	15	7.7

**Table 2 behavsci-12-00373-t002:** Results of the reliability and validity tests.

Item	Construct	St. Estimate	SE	CR	AVE	CR	Cronbach’s Alpha
M_SO1	Manager’s service orientation	0.874	-	-	0.745	0.938	0.929
M_SO2		0.930	0.058	17.702			
M_SO3		0.844	0.064	15.354			
M_SO4		0.800	0.064	13.933			
E_SO1	Sales employee’s service orientation	0.782	-	-	0.763	0.945	0.927
E_SO2		0.850	0.088	13.288			
E_SO3		0.946	0.083	15.223			
E_SO4		0.908	0.085	14.512			
E_CO1	Sales employee’s customer orientation	0.732	-	-	0.529	0.921	0.860
E_CO2		0.677	0.106	10.575			
E_CO3		0.685	0.123	8.376			
E_CO4		0.810	0.131	8.882			
E_CO5		0.725	0.120	8.061			
C_SP1	Service performance	0.870	-	-	0.806	0.960	0.948
C_SP2		0.888	0.042	25.266			
C_SP3		0.919	0.060	18.212			
C_SP4		0.914	0.058	18.098			
C_AUT1	Perceived authenticity	0.771	-	-	0.794	0.921	0.914
C_AUT2		0.940	0.082	14.836			
C_AUT3		0.950	0.079	14.981			
C_LYL1	Customer loyalty	0.951	-	-	0.892	0.960	0.965
C_LYL2		0.939	0.041	24.813			
C_LYL3		0.943	0.043	24.410			

Notes: SE = standard error; CR = composite reliability; AVE = average variance extracted. Model fit: χ^2^ = 292.253; degrees of freedom (df) = 207; χ^2^/df = 1.412; root mean square error of approximation = 0.046; goodness-of-fit index = 0.886; normal fit index = 0.933; relative fit index = 0.918; incremental fit index = 0.979; Tucker–Lewis index = 0.975; comparative fit index = 0.979.

**Table 3 behavsci-12-00373-t003:** Results of the correlation analysis.

Construct	M	St. d	(1)	(2)	(3)	(4)	(5)	(6)	(7)
Manager’s service orientation (1)	6.294	1.094	1	0.280 **	0.131	0.245 **	0.262 **	0.178 *	0.163 *
Sales employee’s service orientation (2)	5.711	1.108		1	0.365	0.303 **	0.242 **	0.216 **	0.119
Sales employee’s customer orientation (3)	6.279	0.772			1	0.288 **	0.222 **	0.153 *	0.195 **
Service performance (4)	4.445	0.776				1	0.639 **	0.622 **	0.116
Perceived authenticity (5)	5.776	0.908					1	0.708 **	0.195 **
Customer’s loyalty (6)	6.085	0.926						1	0.188 **
Customer’s shop transaction periods (7)	3.052	3.579							1

** *p* < 0.01; * *p* < 0.05.

**Table 4 behavsci-12-00373-t004:** Results of the hypothesis testing.

Hypothesis	Path	St. Estimate	SE	CR	Result
H1(+)	Manager’s service orientation → Sales employee’s service orientation	0.166 *	0.051	2.018	Supported
H2(+)	Manager’s service orientation → Sales employee’s customer orientation	0.297 **	0.065	3.867	Supported
H3(+)	Sales employee’s service orientation → Perceived authenticity	0.194 **	0.087	2.575	Supported
H4(+)	Sales employee’s customer orientation → Perceived authenticity	0.191 *	0.128	2.385	Supported
H5(+)	Perceived authenticity → Service performance	0.665 **	0.071	8.727	Supported
H6(+)	Perceived authenticity → Customer loyalty	0.753 **	0.082	10.417	Supported
Control	Customer’s shop transaction periods → Service performance	0.049	0.012	0.949	N/A
	Customer’s shop transaction periods → Customer loyalty	−0.008	0.011	−0.145	N/A

Notes: SE = standard error; CR = composite reliability. Model fit: χ^2^ = 368.699; degrees of freedom (df) = 237; χ^2^/df = 1.556; root mean square error of approximation = 0.0054; goodness-of-fit index = 0.864; normal fit index = 0.913; relative fit index = 0.899; incremental fit index = 0.967; Tucker–Lewis index = 0.961; comparative fit index = 0.967. ** *p* < 0.01; * *p* < 0.05.

## Data Availability

The data are contained within the article.

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
