# Peer review of "Service Orientation and Customer Performance: Triad Perspectives of Sales Managers, Sales Employees, and Customers"

_behavsci, 2022, doi:10.3390/bs12100373_

Round 1

Reviewer 1 Report

Thank you for the opportunity to review this paper. This study explored how sales managers’ service orientation is transferred to employees and how this affects customer attitudes. However, there are several severe problems, which are discussed below:

The authors need to review related literature more thoroughly to demonstrate what research gaps should be filled based on previous research. 

It is necessary to refer to the literature on the relationships among variables, otherwise, their arguments are weak. For example, there are papers on the relationship between service orientation and customer orientation. In addition, is there any reason the perceived authenticity is chosen for the mediator? How about the literature on the transfer hypotheses?  

I wonder why the authors only adopted the managers' service orientation and did not include the transfer of customer orientation.

It would be desirable for the authors to provide more detailed information on how many customers participated and how the responses were transformed into variables and matched with the other variables.

Please provide a correlation table.

Lastly, the degree of similarity between this manuscript and existing studies seems rather high.

I encourage the authors to reconstruct the model, rewrite the hypotheses development to suggest more theoretical implications, and beef up the method contents. 

I hope I have helped the authors with some insights for their future research efforts and wish them good luck.

Author Response

Dear Reviewer, 1:

Thank you for reviewing our paper. After reviewing the related literature in more detail and based on your comments, we have made relevant changes to the manuscript. These changes have been highlighted in yellow for easier referencing.

there are papers on the relationship between service orientation and customer orientation. In addition, is there any reason the perceived authenticity is chosen for the mediator? How about the literature on the transfer hypotheses?  I wonder why the authors only adopted the managers' service orientation and did not include the transfer of customer orientation.

It would be desirable for the authors to provide more detailed information on how many customers participated and how the responses were transformed into variables and matched with the other variables.

Response: The reason for choosing perceived authenticity as a mediator is that the studies on authenticity have hitherto mainly used variables measured from the salespersons’ viewpoint, while we consider the consumer's viewpoint and reveal the effect of authenticity on performance.

The reason for choosing service orientation is that we hypothesize that the manager's individual level of service orientation will affect an individual salesperson's customer and service orientation.

The literature tends to recommend a study design different from that employed here to examine the associations suggested in the current research model. Notably, most existing studies that assess how customer orientation influences customer experience employ the same population sample. That is, they use only employee or customer responses. Consequently, the methodological input of the current study provides further insight into the possible correlation via face-to-face interviews with sales employees and customers through judgment sampling, thereby reducing the standard method bias evident in previous studies on this topic.

Pre-existing literature has generally focused on individual sales employees’ service capacity and attitudes and not on how institutional-level capacity and attitudes are transferred to individual employees.

Please provide a correlation table.

Lastly, the degree of similarity between this manuscript and existing studies seems rather high.

I encourage the authors to reconstruct the model, rewrite the hypotheses development to suggest more theoretical implications, and beef up the method contents. 

Response: The response variable and the correlation table were modified as follows:  Since we need three response sources, this study required the cooperation the department store headquarters, shop manager, sales employees, and their customers. To test the research hypotheses, we surveyed shop managers, sales employees, and their customers from five department store branches in Seoul, South Korea. We conducted face-to-face interviews with sales employees and customers through judgment sampling (i.e., purposive sampling), a non-probability sampling technique in which responses are chosen based on the researcher’s judgment. To measure the relationship between managers and employees, the survey included shop managers and sales employees working at the same shop. Therefore, with the cooperation of the department store headquarters, shop managers were asked to respond to the survey by considering the company (manufacturer) that dispatched them, while the salesperson was asked to answer keeping in mind the company (manufacturer or HR dispatching company) to which they belonged. Moreover, customers were asked to rate the employee–customer relationship on the spot. The interviewer randomly selected two customers who visited each shop in the department store, guided them to a separate location, and conducted the survey; the average of the responses to each question was used. The sample shops included those selling men’s and women’s fashion wear, golf and outdoor goods, and cosmetics. In short, the survey was conducted with a shop manager, a sales employee, and two customers per shop and targeted 200 shops in department stores (see Table I for demographic details).

Classification

N

%

Classification

N

%

Department store branch

Trade Center

39

20.1

Product category

Golf and outdoor

48

24.7

Mokdong

39

20.1

Cosmetics

48

24.7

Pankyo

37

19.1

Men’s fashion

48

24.7

Cheonho

40

20.6

Women’s fashion

50

25.8

Shinchon

39

20.1

Manager’s work experience

Under 3 years

99

51.0

Manager’s gender

Man

49

25.3

3–6 years

36

18.6

Woman

145

74.7

6–9 years

17

8.8

Sales employee’s gender

Man

41

21.1

Over 9 years

42

21.6

Woman

153

78.8

Department store transaction periods of customers

Under 3 years

83

42.8

Customer’s gender

Man

87

22.4

3–6 years

45

23.2

Woman

301

77.6

6–9 years

30

15.5

Sales employee’s work experience

Under 3 years

137

70.6

Over 9 years

36

18.6

3–6 years

40

20.6

Shop transaction periods of customers

Under 3 years

125

64.4

3–6 years

34

17.5

Over 6 years

17

8.8

6–9 years

20

10.3

Over 9 years

15

7.7

Furthermore, the customer survey was conducted in a location away from the shop with the cooperation of the department store’s head office.

We sincerely thank you for evaluating our manuscript

Reviewer 2 Report

First of all, I would like to congratulate the authors. It is an interesting and well written work.
I have no important suggestions. I think it would be useful to include in the abstract some details about the sample and also the period in which the field research took place (maybe in the methodology section).

Author Response

We sincerely thank you for evaluating our manuscript. we strived to improve the details and sophistication of our paper based on your suggestions. Thank you again for taking the time to review our manuscript.

Reviewer 3 Report

Dear authors,

Thank you very much for giving me the opportunity to read and review your paper. In my opinion, the overall topic is of relevance. Getting insights into how institutional-level capacity and attitudes influence employees within organizations  and how this, in turn, affects service performance is an interesting topic. However, I have some remarks and open questions regarding the structure and the methods..

1. Introduction part

I am wondering about the structure of your introduction. In my opinion, you include all relevant aspects of an introduction (see introduction formula), but there does not seem to be a clear structure. In line 69 to 76 you talk about the objective of the study, in line 101 and 110 you start again with the research questions etc. That is really confusing for the reader.  I do not really understand what your main aim is. I would prefer to a more straightforward structure such as:

Problem background, hock, telling that this paper relates to something interesting. What makes your topic so interesting? Prior work in order to understand the contribution of your paper. Value added and some kind of roadmap.

Your literature review seams a little arbitrary and I cannot see how you came up with the studies you cite? It seems a little random and I miss other studies that included the big five factor model in the context of meat consumption and animals. .

2. Materials and methods and 3. Research Model and Hypotheses

You directly start with the explanation of your constructs in chapter 2. One sentence that gives the reader some hint what will follow in the chapter would be nice, either as a roadmap at the end of chapter 1 or directly at the beginning of chapter 2.

I don’t think the title of the chapter is correct. Isn’t it more a literature overview what you resent? You say nothing about the materials an methods here?

I would recommend to collapse chapter 2 and 3 name it something like “Literature overview and derivation of hypotheses.” This seems to be smore precise. I am wondering about the formulation of your hypotheses: Why do you “only” talk about correlations? Why you did not formulate the hypotheses as e.g. follows: Sales managers’ service orientation positively influences sales employees’ service orientation, as it is shown in your hypothetical model? This hold true for all hypothesis.

I would recommend to always use the same wording of the constructs: You use “Perceived authenticity” in the model but in H3: customers’ perception of authenticity.”  Therefore, either always use customers’ perceived authenticity or customers’ perception of authenticity. Otherwise it is confusing. The same holds true for all other constructs.

4. Research methods and analysis

Open a new chapter 3 “Material and methods”. Here I would include all information form line 332 to probably line 374. Additionally, a chapter 4 Results is needed. All information starting from line 375 would belong to the results section.

How did you came up with your sample size? I am puzzled about the five factor model. Who came up wit this concern you mention in line 399. I assume another reviewer? However, I would not formulate it like that. This is confusing. It is very good that you address the issue with the high correlation between the construct. But, besides arguing from a statistical perspective you can argue content wise. Anyway, please reformulate this paragraph (397-410). It is confusing what belongs to the five factor and what to the six factor model. Can you provide more information on how you perform the chi-square difference test?

Is it possible to support a hypothesis or do you just reject H0?

I am really puzzled about customer’s shop transaction periods (7)? Where does this come from (table 3, 4) Where does this control variable come from (line 431). It seems to come out of nowhere? Please explain that in more detail.

5. Discussion

I would be more careful in formulation your managerial implications, especially focusing on your limitations.

GENERAL REMARKS:

You use very often the same words “Additionally” (line 69, 81).I would recommend to send the document to an English proofreading service.

I think this is an interesting piece of work that has the potential to contribute to literature. However, the elaboration and details are still lacking and need to be improved. I am looking forward to the revisison.

Author Response

Dear Reviewer, 3:

We sincerely thank you for taking the time to carefully evaluate our manuscript. We made changes based on your comments, which are highlighted in yellow in the revised paper.

  1. Introduction part

What makes our topic interesting is that the shop manager's service orientation is transferred to the salesperson's service orientation. Additionally, we found that the service and customer orientation of the salesperson have an important effect on the evaluation of the perceived authenticity of the customer toward the salesperson. In other words, good service and providing the right service for customers are two different things, but both have an important impact on the authenticity evaluation. Further, the evaluation of authenticity affects service performance and customer loyalty. This text is given on page 3, from lines 104 to 114 (Accordingly, the following research questions are raised: 1. does the service orientation of department store managers affect the individual-level service and customer orientation of sales employees? 2. Do sales employees’ service and customer orientation positively impact customers to perceive service authenticity, thus resulting in effective service performance and customer loyalty?).

  1. Materials and methods and 3. Research Model and Hypotheses

Thank you for your suggestions here, and the relevant text was modified accordingly (The remainder of this paper is structured as follows. Section 2 provides an overview of the relevant literature regarding the constructs used in this study: service orientation, customer orientation, perceived authenticity, customer performance, and customer loyalty. Furthermore, the hypotheses proposed in this study are described. Section 3 presents the materials and methods and the findings, and Section 4 concludes with a discussion of their implications.).

Yes, and we will change “2. Literature Overview” to “2. Literature Overview and Derivation of Hypotheses”.

To reduce confusion, we changed for all other constructs (H3: Sales employees’ service orientation is positively correlated with customers’ perceptions of authenticity. H4: Sales employees’ customer orientation is positively correlated with customers’ perceptions of authenticity.).

The content you refer to has been corrected and highlighted.

  1. Research methods and analysis

We appreciate the reviewer recognition of these aspects of our manuscript. The content you mentioned has been edited and highlighted. Thank you for the advice regarding this section.

Since we need three response sources, this study required the cooperation the department store headquarters, sales manager, sales employees, and their customers. To test the research hypotheses, we surveyed sales managers, sales employees, and their customers from five department store branches in Seoul, South Korea. We conducted face-to-face interviews with sales employees and customers through judgment sampling (i.e., purposive sampling), a non-probability sampling technique in which responses are chosen based on the researcher’s judgment. To measure the relationship between managers and employees, the survey included sales managers and sales employees working at the same shop. Therefore, with the cooperation of the department store headquarters, sales managers were asked to respond to the survey by considering the company (manufacturer) that dispatched them, while the sales employee was asked to answer keeping in mind the company (manufacturer or HR dispatching company) to which they belonged. Moreover, customers were asked to rate the employee–customer relationship on the spot. The interviewer randomly selected two customers who visited each shop in the department store, guided them to a separate location, and conducted the survey; the average of the responses to each question was used. The sample shops included those selling men’s and women’s fashion wear, golf and outdoor goods, and cosmetics. In short, the survey was conducted with a shop manager, a sales employee, and two customers per shop and targeted 200 shops in department stores (see Table I for demographic details). Furthermore, the customer survey was conducted in a location away from the shop with the cooperation of the department store’s head office.

We first examined the fit indices of the proposed six-factor model (i.e., managers’ service orientation, sales employees’ service orientation, sales employees’ customer orientation, perceived authenticity, service performance, and customer loyalty).

Also, Thank you for your question. We used this for our footnotes. The text is given on page 12, line 488. If the test statistic follows a chi-square distribution, hypothesis testing can be performed. In this case, the following holds: 1) the null hypothesis is rejected (and the alternative hypothesis is accepted) if it is difficult for the test statistic to follow the chi square distribution; 2) if the test statistic can sufficiently occur x, the null hypothesis is rejected, and the confidence level or P value is used to determine whether an event is probable or unlikely.(The chi-square difference tests revealed that the six-factor model yielded a better fit than the five-factor model (Δχ2(5) = 163.592, p<0.01).

  1. Discussion

Thank you for pointing this out. we have edited this section and removed repetition as much as possible.As per your suggestion, we have also used a professional editing service.

The revised content is mentioned on pages 14 to 15, from lines 577 to 633.

(The current study’s empirical findings provide practical and managerial implications for sales organizations, especially large retailers and department stores. An appropriate education and compensation system should be introduced to effectively manage customer orientation. An atmosphere allowing active management support and the free exchange of opinions should be provided. The study suggests adopting a method based on organizational goals and objectives to achieve customer performance.

First, the findings highlight why managers should cultivate service orientation: a manager’s service orientation impacts sales employees and, ultimately, customer performance. Sales managers are expected to increase the sales employee’s service orientation by strengthening workplace relationships. In role-based relationships, sales managers need to communicate in the sales process, emphasizing the goal of the sales organization, that is, service orientation, and the virtues that sales employees are expected to possess. In personal workplace relationships, sales managers become role models so that the sales employees can voluntarily show service-oriented behaviors in the sales process. In order to do this, sales managers need not only communication to simply communicate as role occupants to accomplish work tasks [16,138], but also share a unique experience with sales employees to secure emotional ties or personal intimacy. Moreover, Individual service orientation can be disseminated as organizational culture because lower-level employees interact frequently and directly with managers. Service organizations should pay specific attention to and invest in providing customer service-related training to instill awareness on the importance of service-provisioning attitudes and behaviors in store managers and employees to ensure their desired service performance. Therefore, sales organizations need to support service orientation at the firm level.

Second, this study demonstrates the top-to-bottom transferability of service attitudes in practice, allowing for better customer performance in a department store. Examining the transfer of service attitudes can provide managers with valuable insights into designing training mechanisms that instill service attitudes in employees at different levels. This is especially important in South Korea, as product category managers are employed by department stores, whereas manufacturers or tenants appoint sales managers and sales employees. In some cases, sales employees are hired by the manufacturers, but there are also cases where they are hired by an HR outsourcing company, making it difficult for the sales managers to directly train them. Under this fragmented human resource management structure, it is difficult to provide comprehensive sales and service training or sales instruction. Therefore, from the manufacturer’s perspective, it is more appropriate for sales managers to provide implicit rather than explicit training to inculcate service attitude in employees, and as we mentioned above, strengthening the personal workplace relationship between sales managers and sales employees can be an effective alternative.

Third, sales organizations must instill service and customer orientation to achieve their service objectives and goals. This study reveals that service and customer orientation positively affect service performance. Previous studies have shown that sales employees’ service and customer orientation can be enhanced by their positive attributes [139] and organizational support [140,141].

Fourth, sales organizations should ensure that their employees have perceived authenticity to maintain customer loyalty. As illustrated, businesses can only obtain customer loyalty when perceived authenticity is guaranteed. According to Grandey et al. [132], customers perceive authenticity in sales employees’ attitudes based on verbal and nonverbal communication. Therefore, sales organizations should consider the importance of gestures, eye contact, tone, facial expressions, and verbal communication, and train employees to deliver authenticity during sales encounters.

Finally, to establish a service-oriented organizational culture and develop a sincere workforce, managers should focus on developing relationships with their sales employees. Establishing an organizational culture that faithfully preserves the essence and form of attachment to the work and organization is important. This suggests that customer-oriented strategies should be established, keeping in mind employee satisfaction and emotional attachment to the organization. In other words, to raise the level of customer service, it is necessary to establish a desirable organizational culture and develop human resources and internal marketing policies that increase organizational attachment. An organizational culture that focuses on building relationships rather than just performance induces strong employee attachment to the job and organization, positively affecting performance through voluntary efforts to improve customer experience.).

Overall, we strived to improve the details and sophistication of our paper based on your suggestions. Thank you again for taking the time to review our manuscript.

Round 2

Reviewer 3 Report

Dear authors,

Thank you very much for carefully considering all my comments!

I am looking forward to the published article.

All the best

Author Response

Thanks for your positive comments!